# The Utility of Calibrating Wearable Sensors before Quantifying Infant Leg Movements

**DOI:** 10.3390/s24175736

**Published:** 2024-09-04

**Authors:** Jinseok Oh, Gerald E. Loeb, Beth A. Smith

**Affiliations:** 1Division of Developmental-Behavioral Pediatrics, Children’s Hospital Los Angeles, Los Angeles, CA 90027, USA; joh@chla.usc.edu; 2Alfred E. Mann Department of Biomedical Engineering, Viterbi School of Engineering, University of Southern California, Los Angeles, CA 90089, USA; gloeb@usc.edu; 3Department of Pediatrics, Keck School of Medicine, University of Southern California, Los Angeles, CA 90033, USA; 4Developmental Neuroscience and Neurogenetics Program, The Saban Research Institute, Children’s Hospital Los Angeles, Los Angeles, CA 90027, USA

**Keywords:** calibration, movement quantification, offset error, inertial measurement unit (IMU), wearable sensors, reproducibility

## Abstract

While interest in using wearable sensors to measure infant leg movement is increasing, attention should be paid to the characteristics of the sensors. Specifically, offset error in the measurement of gravitational acceleration (*g*) is common among commercially available sensors. In this brief report, we demonstrate how we measured the offset and other errors in three different off-the-shelf wearable sensors available to professionals and how they affected a threshold-based movement detection algorithm for the quantification of infant leg movement. We describe how to calibrate and correct for these offsets and how conducting this improves the reproducibility of results across sensors.

## 1. Introduction

The consistent quantification of infant leg movements has potential clinical applications. For example, some researchers found a significant correlation between the frequency of kicking movements in infancy and the onset of walking [1]. Others have shown that leg movement characteristics are altered in infants born preterm [2], with low birth weight [3], or with Down’s syndrome [1,4]. These findings suggest the diagnostic potential of infant leg movement kinematics, particularly when measured at a very early age.

While infant leg movements can be studied with three-dimensional motion capture systems, these systems are only useful for measuring movement over a short period (minutes) of observation. The same is true of video-based movement analysis. Because it currently requires visual annotation by experienced coders, in our experience, analyzing 5 min of infant leg movement data takes a few hours. In both cases, quantification is thus limited to minutes of behavior and does not capture the full range of behaviors that infants produce across one or more days. Alternatively, interest is growing in measuring the full-day movements of infants in their natural environments using wearable sensors [5,6,7]. Commercially available wearable sensors are relatively inexpensive and easy to operate, and some sensors can even be shipped to the homes of participants. With the currently available battery life, such sensors can record infant leg movements continuously for days. It is important to note, however, that while collecting wearable sensor data with off-the-shelf sensors is relatively easy, analyzing the data is not. Sensor-based metrics need to be created and validated for infants, as off-the-shelf analyses are not available for analyzing infant leg movements.

Any measurement technique requires decisions about what aspects of behavior are useful to measure and what methods for data analysis are needed to quantify those aspects. Movement duration is one such aspect that may help distinguish infant-generated movements from any other movement such as cuddling of caregivers or vibrations from different sources. A previous study reported that infant spontaneous kicks are typically less than 500 milliseconds long [8]. This translates to some number of data points depending on the sampling frequency (*f_s_*) of the sensor. Movement intensity is another aspect to consider. Commercially available inertial measurement units (IMUs) generally transduce and quantify all three axes of translational acceleration plus all three axes of gyroscopic rotational velocity. Infant leg movements can be defined according to a minimal acceleration magnitude and counted according to the number of times the magnitude crosses a predefined threshold [7].

Researchers interested in using wearable sensors to measure the frequency of infant leg movements and estimate physical activity intensity, particularly those newly introduced to the methodology, need to understand the kinds of errors to which multi-axis accelerometers are prone and how they might affect the results of movement detection and counting algorithms. This knowledge will facilitate not only the collection but also the comparison of data from multiple studies and IMU devices. Specifically, errors of gain (i.e., the static effects of gravity reported are proportional to but not identical with 1 *g* = 9.8 m/s^2^), non-orthogonality of axes (i.e., there is no orientation in which all the effects of gravity appear on only one axis), offsets (i.e., when oriented orthogonally to gravity, the signal is non-zero), and noise (fluctuations around the mean when the sensor is at rest) need to be considered.

This study aims to raise awareness on the need for calibration adjustments for IMU data analysis by (1) reporting the gain, misalignment (non-orthogonality of axes), offset, and noise of measurement axes for IMUs used to collect movement data, (2) providing a tutorial on a simple calibration and correction method [9], which has been noted by researchers over time [10,11], and (3) demonstrating its ability to improve the comparability of movement counts among the different IMUs. Three samples from each of three different models of IMUs for each of these types of error were tested. The results demonstrate that only offset errors were significant and that they had significant effects on movement detection, which were mitigated by the calibration procedure described herein.

## 2. Background

Identifying discrete, countable movements from continuous analog signals generated by wearable sensors necessarily requires threshold-based algorithms such as those developed and validated previously [7] and that employed in this study. Briefly, the time-stamped Euclidian norm of acceleration (∥a∥) measured along three axes is calculated (square root of the sum of the squares of each axis of acceleration). Translational accelerometers respond to the static effects of gravity, so ∥a∥ is detrended ($∥a{∥}_{d}$) by subtracting its median (based on the assumption that the infant is at rest at least 50% of the time) so that the values of the detrended norm are centered around the baseline of 0 *g*. Positive and negative threshold values are dynamically set near ±0.1 *g* to detect movements that start with or against the static effects of gravity, respectively. A movement starts with a value of $∥a{∥}_{d}$ above the positive threshold or below the negative threshold. The movement ends after consecutive values cross the baseline twice from different directions within 1.5 s (indicating a movement that is accelerated from and decelerated to a resting posture), with at least one of the values crossing the threshold opposite to the one already crossed. During the entire duration of the movement, the angular velocity norm with its median subtracted should be greater than 0. The portion of $∥a{∥}_{d}$ (connected pink dots) marked by a green line in Figure 1B showcases the described movement.

A movement may be missed when accelerometer analyses are not adjusted for calibration. Figure 1A showcases where uncalibrated infant leg movement data are provided to the algorithm. The $∥a{∥}_{d}$ values during this 1.6 s period are shifted upward from the baseline of 0 *g* (red solid line) because of the uncorrected errors in the offset. The trajectory of $∥a{∥}_{d}$ rises and crosses the positive threshold but does not cross the negative threshold. The algorithm thus decides that this trajectory is not a movement. When errors are corrected, the missed movement is counted (Figure 1B). For asymmetrical movements, failure to correct offset errors can also result in the false-positive identification of movements.

## 3. Materials and Methods

### 3.1. Selected Wearable Sensors

Three tri-axial IMU models in common use were tested to compare the presence of gain and offset error: Opal version 2 (APDM Inc., Portland, OR, USA), Ax6 (Axivity Ltd., Newcastle, UK), and Movesense Active HR2 (Movesense Ltd., Vantaa, Finland). Opal has been a popular choice that has been validated for recording various aspects of human movement [12,13,14] and even used as a reference when validating other sensors [15]. The Ax6 and Active sensors are also often used to measure or recognize human movement [15,16,17,18,19].

Specifications of the sensors are provided in Table 1. The sensors had distinct shapes. Opal and Ax6 were cuboids with round corners, having orthogonal faces aligned with the axis measuring *g* (gravity axis). In contrast, Active had a button-like shape with top and bottom surfaces well aligned with gravity. The other two axes were found to have limited accuracy, because the orientation of the axes was not indicated on the circular shape of the sensors.

### 3.2. Preparation of Datasets

Two types of datasets were prepared. Calibration datasets were generated from 60 s recordings of each of the three IMU models. Each sensor was placed on a level floor in two different orientations (up and down) per axis for 10 s (10 × 6 = 60 s of recording per sensor). This allowed one of the three axes to be parallel to the direction of gravity and measure *g* along that specific axis. These measurements were later compared with the expected *g* value (−1 or +1) to estimate the gain and offset [9,20]. Three calibration datasets were prepared for each of the three IMU models using three different sensors of each model. An iOS application, *Movesense Showcase* (Suunto Oy, version 1.1.0), was used to find the approximate x and y axes when preparing the calibration dataset of the Active sensors. The sensors were connected through Bluetooth, and the app streamed the *g* value measured along the axes. While observing the streamed values, the sensor was placed in orientations that maximized the measured *g* (−1 or 1) along a single axis (x or y) while minimizing the amount along the other two axes (i.e., measuring values close to 0 *g*).

A movement dataset was prepared from 5 min of recording using all three sensors wrapped together and placed on the right dorsal forearm of an adult (JO). The specific location of the sensor placement was 5 cm from the wrist (a line connecting the styloid processes of the ulnar and radius bones). The combined mass and bulk of all three sensors precludes carrying this out on an infant limb. The sensors recorded 200 linear forearm movements (100 thrusts and 100 pull-backs) that mimicked the movement of the lower leg during typical supine kicking [21] by an infant. Supine kicking is one of the earliest and most prominent movements observed among infants between 0–6 months. It involves the synchronized movements (flexion or extension) of an infant’s hip and knee joint. In previous studies, leg movements including kicking were recorded using wearable sensors placed near the ankles of infant legs [7]. To imitate kicking recorded by sensors at the ankle, sensors were placed near the wrist of the adult, and a movement started with the shoulder abducted at around 45°. The internal rotation of the shoulder joint and the extension of the elbow joint (thrust) imitated an infant’s extension of the hip and the knee joints. The external rotation of the shoulder and the flexion of the elbow (pull-back) corresponded to the synchronized flexion of the joints of an infant. During movements, the arm was in the air, not touching any surface.

### 3.3. Estimation of Errors and Measurement Noise

Prior to gain estimation, offsets for each axis were determined and corrected. If a non-gravity axis produced the same non-zero value when the IMU was flipped in the gravity axis, then that was indicative of a simple offset. If the non-zero values were different in the two orientations, this was indicative of an alignment error. Misalignment is not important for movement detection because it is based on the non-dimensional norm, but it must be considered to determine the magnitude of the offset, which is the mean of these two signed values. Consequently, the mean of each axis’ values when it was a non-gravity axis (expected value: 0 *g*) was the axis offset. Axis misalignment was the mean of the absolute values of the axis after the axis offset was removed. Finally, offset-removed measurements of ±1 *g* were arranged. Windowed mean of values near 1 *g* whose standard deviation was less than 0.01 represented a 1 *g* measurement of the axis (*m*_1*g*_). The same was true for values near −1 *g* (*m*_−1g_). Ultimately, the gain (*G_axis_*) of an axis was estimated using the following equation: *G_axis_* = (*m*_1*g*_) – (*m*_−1*g*_)/2.

Measurement noise, the spread around the mean of a steady-state signal, was also measured along each axis. A noisy sensor will automatically generate a larger acceleration norm and possibly more movement counts than a noise-free sensor. To examine if this was true with our algorithm, the measurement noise of each axis was defined as twice the standard deviation of the values used to derive *m*_1*g*_.

### 3.4. Error-Correction Procedure

The movement dataset for each sensor was first corrected for the offset of the corresponding sensor. The axis-specific offset error was subtracted from the recorded linear acceleration values of the matching measurement axis. The values were then each divided by the gain value of the corresponding axis. Misalignment error and measurement noise were not addressed. The former was not considered because it did not have any influence on the movement detection algorithm based on the dimensionless norm. The latter was not corrected based on a post hoc assessment of its influence on the work of the algorithm.

### 3.5. Filtering High Frequency Components

The three different IMU models had different sampling rates, reflective of the different frequency responses available from their sensors. To be certain that the same movement parameters were detected by all three, it was necessary to filter out the higher-frequency components of the datasets from Ax6 and Active to be equivalent to the lower-frequency Opal. A first-order Finite Impulse Response (FIR) low-pass filter with the Hamming window and a cut-off frequency set at 8 Hz was applied to all three sensor types. This was below the Nyquist frequency of the Opal sensor (10 Hz) and above the maximum frequency of the recorded movements (~4 Hz). The digital filter was designed using the *firwin* function of Python’s SciPy module (Python 3.10.6; SciPy 1.11.0).

## 4. Results

### 4.1. Offset, Misalignment, Gain, and Noise of Measurement Axes

All three sensors exhibited varying levels of offset, misalignment, and gain errors and noise. When the sensors were in various orientations, the acceleration in the non-gravitational *Y* axis was not consistently zero, as depicted in Figure 2. When the *X* axis was the gravity axis, measured *g* values (solid lines) were near −0.07, even when the sensor was flipped. When the *Z* axis was the gravity axis, one orientation returned Y values near −0.065, while the other generated values near −0.095. This is indicative of axis misalignment as well as offset. The calculated offset and misalignment of the axis were −0.074 *g* and 0.016 *g*. Table 2 reports different types of errors as well as the noise of the measurement per axis. In general, the Ax6 sensors showed greater offset values than the other two sensors, with all three sensors having non-zero axis offset values.

After the offset error correction, the gain values of the different sensors were near 1. The gains of the Opal sensors were the closest to 1 (average of the three measurements were 0.999, 0.999, and 1.000 for the *X*, *Y*, and *Z* axes, respectively), but those of the other sensors were also comparable. The measurement noise was the smallest for the Opal sensors (<0.003 *g*) but was deemed negligible for all the axes of all the sensors (<0.009 *g*).

### 4.2. Effect of Calibration and Low-Pass Filtering on the Algorithm Output of Data from Different Sensors

The forearm movement data collected with the three sensors were preprocessed and provided to our movement detection algorithm. The efficacy of the two preprocessing steps—calibration and low-pass filtering—were assessed according to their improvements on the reproducibility of movement counts across sensor types. Table 3 shows the movement counts from the three sensors’ recordings under different conditions. When the datasets were not processed at all (Raw), the three movement counts differed substantially from one another, ranging from 185 to 269. However, the counts became more similar to each other once the datasets were calibrated and filtered (Opal v2: 195, Ax6: 175, Movesense Active: 213). The Opal sensor’s recording generated the most accurate count of movements; the recordings of the other two sensors either underestimated (Ax6) or overestimated (Active) the number of movements.

## 5. Discussion

The three types of wearable sensors that were tested demonstrated varying levels of measurement errors, including misaligned axes and gain/offset problems. Correcting these errors contributed to generating more comparable results using a threshold-based movement detection algorithm with data from different sensors. This implies that behavioral scientists who consider using commercially available wearable sensors to quantify infant leg movements should inspect the sensors before making use of the raw data. The simple calibration procedure and error-correction methods described herein should correct for the major sources of sensor error.

### 5.1. Offset Error Is One Main Source of Incorrect Estimation

Sensor offset error appears to be the biggest concern in estimating the number of movements using the norm of linear accelerations coming from IMUs. The gains or sensitivity values of the sensors were all near 1. The measurement noise was also less than 0.01 *g*, and most misalignment errors were less than 0.02 *g*. The offset errors, however, were almost always greater than 0.01 *g*, with some values surpassing 0.1 *g*.

How does the untreated offset error influence a movement detection algorithm like the one discussed in this article? When calculating the norm, positive or negative offset errors are also squared and added, making the norm at rest greater than 1 *g*. As a result of axis alignment errors, the measured offsets depend on the sensor orientation, which will change during a recording session, so the detrended norm sits below 0 *g* at rest at some times and above 0 *g* at rest at other times. In the former scenario, norm values corresponding to movements are less likely to cross a positive threshold. In the latter, a negative threshold is not crossed (Figure 1A). Offset errors in a given sensor vary greatly in both magnitude and sign. Any attempt to detect movements based on acceleration in a single axis instead of computing the norm count over/underestimates the true acceleration depending on the sign of the offset and the orientation of the sensor axis with respect to both gravity and the infant’s limb.

Offset error along measurement axes can explain a recent finding that also supports the need for the calibration of wearable sensors in measuring physical activity [22]. Researchers proposed different sensor measures to estimate physical activity (PA) levels measured with oxygen consumption (VO_2_), such as the count data provided through the sensor systems of ActiGraph Llc., Euclidean Norm Minus One (ENMO), or the mean amplitude deviation (MAD) of raw sensor data. Among the measures, the last two are easily calculable for data from any sensor, and distinct values were proposed as thresholds to classify different PA intensities. Weitz et al. [22] investigated the effect of calibration on the estimation of time spent in moderate-to-vigorous physical activity (MVPA) using the two measures. Researchers used low and high thresholds of ENMO [23] and MAD [24] to estimate time spent in MVPA. For a sensor recording of an adult’s daily activity, the sum of the time windows where the ENMO or MAD values are between the two thresholds defining MVPA was equal to the daily time spent in MVPA. Researchers reported that the average of the estimated minutes of MVPA per day based on ENMO was significantly overestimated when the sensor data for analyses were not calibrated. Researchers further illustrated that a post hoc auto-calibration [20] decreased ENMO almost by half and thus reduced the estimated MVPA volume. Varying amounts of offset error make the overall Euclidean norm greater than 1 *g* at rest. Consequently, the chance of ENMO calculated from uncalibrated sensor data crossing the lower threshold of MVPA will be significantly higher than that of the measure coming from calibrated data. Our study and that of Weitz et al. [22] therefore showcase two different examples of how offset error can negatively influence the accuracy and reproducibility of measurements of the frequency of leg movements and intensity of physical activity.

### 5.2. A Comparison with Alternative Calibration Methods

Calibration methods that estimate errors from the actual recording of movements using wearable sensors have been proposed [9,20,25]. Such methods avoid the requirement for a separate calibration and can be applied to pre-existing records for which the source sensor is unavailable. Preparing datasets for a separate calibration process was described as cumbersome in studies with a high throughput [20]. This is true when thousands of sensors are used in a cohort study. Still, preparing a separate dataset of a sensor would take at most 2 min. The benefit of this method is that the datasets are from static moments at known sensor orientations, making the estimation of misalignment and gain errors reliable, while other more automated methods can speculate at best from the periods of recordings estimated to be the times of non-wear or zero-to-minimal movement such as sleeping. Fortunately, the micro-electromechanical systems (MEMS) technology used in modern IMUs appears to perform stably over time [26]. This suggests that in studies that utilize smaller numbers of sensors, experimenters can collect the calibration data once and keep track of which sensors are used to create which datasets so that their raw data can be corrected before further analysis.

### 5.3. Additional Aspect to Be Considered

Although not investigated in this report, one other aspect to consider when using multiple wearable sensors simultaneously to quantify movements is clock synchronization between sensors. For example, if researchers want to answer whether an infant’s two legs moved simultaneously and in-phase or reciprocally, the samples from the sensors on each leg should have synchronized timestamps. Opal sensors do have a hub device that communicates with sensors to synchronize the internal clocks of the sensors. This is not the case for other sensors. One simple solution to this issue could be to have a noticeable event at the beginning and at the end of a recording (e.g., hitting the sensors together five times) and later interpolate the timestamps of the raw data based on the events.

## 6. Limitation

While an adult’s linear arm movements resemble infant supine kicks in how the relevant joint movements are coordinated, they neither represent the full spectrum of the leg movements infants make during a day, nor are they similar in the actual magnitude values associated with the movements. However, the focus of this article is to demonstrate the problem caused by the measurement errors of commercially available wearable sensors when using an acceleration magnitude threshold-based movement detection algorithm to quantify infant movements and how to address the problem independent of sensor choice. Testing the effect of calibration on each sensor’s performance can be carried out by comparing the movement counts derived from uncalibrated and calibrated datasets of the same sensor. The direct comparison of infant movement detection by different sensors is not possible because infant behavior varies between sessions, and it is not feasible to place more than one sensor at a time on an infant’s limb.

## 7. Conclusions

Wearable motion sensors are subject to offset, misalignment, and gain errors and noise. All kinds of errors were present in all three types of sensors tested in this study, with the offset error being the largest in magnitude. The simple calibration and correction method provided here improved the reproducibility of the movement counts obtained by a threshold-based algorithm. Studies that compare data from participants wearing different sensors should obtain calibration data for each sensor and use them to correct the recorded data before quantifying the movements of each participant.

## Figures and Tables

**Figure 1 sensors-24-05736-f001:**
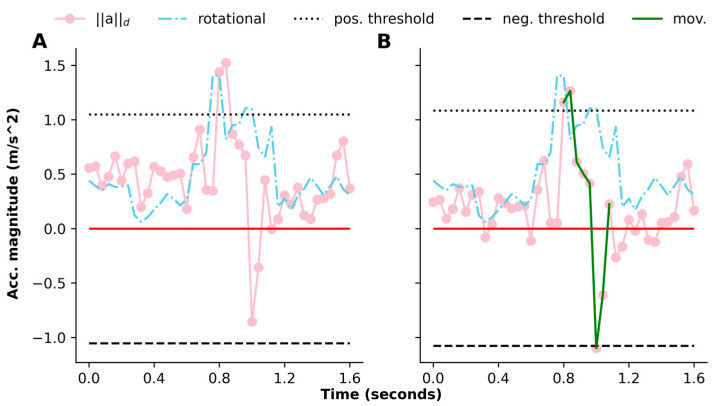
Movement detection algorithm’s output for raw and calibrated wearable data analysis. (**A**) Magnitude of the detrended acceleration norm ($∥a{∥}_{d}$) in a window of 1.6 s when gain and/or offset is not corrected. The positive threshold (pos. threshold) is crossed once, but the negative threshold (neg. threshold) is not crossed. This is because, overall, $∥a{∥}_{d}$ is shifted upward within this window. No movement is defined within this window. (**B**) Same data after offset and gain error corrected; $∥a{∥}_{d}$ values are shifted downward, and the trajectory crosses both the positive and the negative threshold, allowing a movement (mov.) to be detected after calibration. Median subtracted angular velocity norm (rotational) is greater than 0 during the duration of the movement.

**Figure 2 sensors-24-05736-f002:**
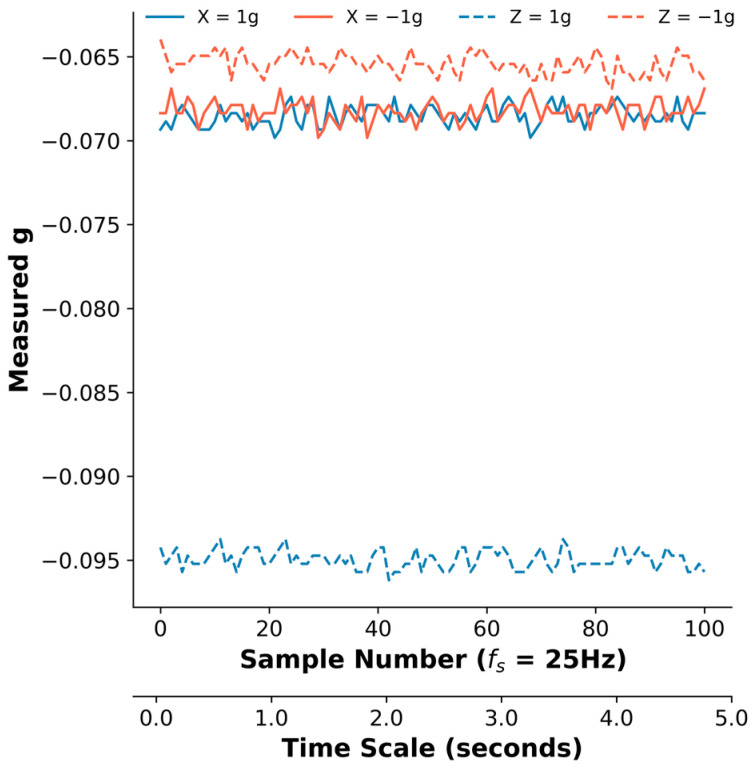
Gravitational acceleration (*g*) values measured along the *Y* axis of an Ax6 sensor. The *Y* axis was a non-gravity axis. Measurement was 5 s long. When the *X* axis was the gravity axis, the *Y* axis consistently measured values near −0.07 *g*, even when the sensor was flipped (solid lines). However, when the *Z* axis was the gravity axis, the Y values (dashed lines) differed substantially, indicating axis misalignment. The offset of the axis was determined by the mean of the measurements under four different orientations. This sensor’s *Y* axis offset was −0.074 *g*.

**Table 1 sensors-24-05736-t001:** Specifications of the wearable sensors.

Name	Dimension (mm) ^1^	Sampling Frequency (Hz) ^2^	Range ^3^	Resolution ^3^	Noise ^3^
Opal V2	43.7 × 39.7 × 13.7 (L × W × H)	20	±16 g (A), ±2000 deg/s (G)	14 bits (A), 16 bits (G)	120 μg/√Hz (A), 0.0025 deg/s/√Hz
Ax6	23 × 32.5 × 8.9 (L × W × H)	25		16 bits (A, G)	N/A
Movesense Active HR2	36.6 × 10.6 (D × H)	52		16 bits (A, G)	N/A

^1^ Opal V2 and Ax6 are cuboid-shaped, while Movesense Active has a button-like shape (L: length, W: width, H: height, D: diameter). ^2^ Frequency used to record data is reported. ^3^ Measures for an accelerometer (A) or a gyroscope (G), respectively; N/A: not provided from data sheets.

**Table 2 sensors-24-05736-t002:** Offset, misalignment, gain, and noise of the three measurement axes of sensors.

Error	Axis	Opal v2 *			Ax6 *			Movesense Active HR2 *		
		1	2	3	1	2	3	1	2	3
Offset	X	0.018	0.013	0.013	0.062	0.030	0.065	−0.002	−0.017	0.084
(ideal = 0)	Y	−0.010	−0.023	0.013	−0.074	−0.067	−0.084	−0.029	−0.083	−0.185
	Z	−0.012	−0.004	0.027	0.030	0.018	−0.007	0.114	0.063	0.144
Misalign.	X	0.016	0.008	0.009	0.011	0.009	0.009	0.005	0.008	0.005
(ideal = 0)	Y	0.006	0.006	0.000	0.016	0.005	0.001	0.004	0.021	0.004
	Z	0.006	0.023	0.014	0.011	0.012	0.004	0.004	0.001	0.017
Gain	X	0.999	0.999	0.999	1.002	0.998	0.998	1.009	0.997	1.008
(ideal = 1)	Y	0.999	0.999	0.999	0.998	0.994	0.993	1.001	1.003	1.004
	Z	1.000	1.000	0.999	1.009	1.011	1.013	1.002	1.005	1.009
Noise	X	0.002	0.002	0.002	0.003	0.005	0.009	0.004	0.005	0.004
(ideal = 0)	Y	0.002	0.002	0.003	0.001	0.001	0.002	0.004	0.004	0.008
	Z	0.002	0.001	0.002	0.002	0.002	0.002	0.004	0.004	0.004

* Three different sensors of each type were used (1, 2, and 3). The unit of offset, misalignment, and noise is *g*.

**Table 3 sensors-24-05736-t003:** Movement counts with and without preprocessing steps (reference count: 200).

Sensor	Raw	C	C + F	Noise after C + F (*g*)		
				X	Y	Z
Opal v2	269	269	195	0.001	0.001	0.001
Ax6	185	196	175	0.001	0.001	0.002
Movesense Active HR2	231	230	213	0.003	0.003	0.003

C: Calibrated gain and offset errors, F: filtered with a first-order low-pass filter (cut-off frequency = 8 Hz).

## Data Availability

The data presented in this study are available on request from the corresponding author (B.A.S.) as a Data Use Agreement is necessary.

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
