# Peer review of "The Utility of Calibrating Wearable Sensors before Quantifying Infant Leg Movements"

_sensors, 2024, doi:10.3390/s24175736_

Round 1

Reviewer 1 Report

Comments and Suggestions for Authors

Ms. Ref. No.: sensors-3190587

Title: Wearable Sensors Should be Calibrated Before Quantifying Infant Leg Movements

Authors: Jinseok Oh, Gerald E. Loeb, Beth A. Smith

Journals: Sensors

Reviewer’s Comments:

The article reports on the calibration technique of three different off-the-shelf wearable sensors to monitor infant leg movement. Authors demonstrated offset correction methods and reproducibility of the result across sensors through calibration. The paper is well written and understandable, although in some parts authors used first-person pronouns (e.g. we, our etc.). It will be appropriate to avoid first-person pronouns in article. Change the language from first-person pronouns to passive voice. Some other revision suggestions are as follows,

1.      Title of the article requires modification as it sounds vague. Current title seems like a statement.  It is, normally, a well-known practice to calibrate the wearable sensors prior to use.

2.      In Abstract section, last two lines sounds repetitive. Authors should modify that part.

3.      In Page 5 and Line 206, check the sentence “measurement per each axis”.

4.      Does the mounting position of the sensor on the body affects the calibration gain and offset error measurement.

5.      Table 3 results are not conclusive. Especially for Ax6 sensor result become worsen after calibration and filtered.

                 I.           Why authors choose only 200 counts? What if they go for 1000 or 5000 counts? Whether the proximity of the count results for all three sensors will be similar? How repeatable or reproducible the results will be for higher counts?

               II.           For Opal v2 and Active HR2, only calibration does not have any affects. But after offset error count results moves close to the actual reading.

            III.           Whereas, for Ax6 sensor error increases from 7.5% for 185 counts (raw) to 12.5% after C+F modification. So, the question remains: what type of accuracy authors are looking from the sensors? What is the level of tolerance (error in %) for this measurement?

            IV.           Can authors demonstrate results for actual measurement of the sensors on infant leg movement after calibration and off-set error correction. And then compare the sensors results with the established state-off-art measurement tool/ method like, three-dimensional motion capture systems or video-based movement analysis. This will properly demonstrate if the method really works in operation!

Comments on the Quality of English Language

The paper is well written and understandable, although in some parts authors used first-person pronouns (e.g. we, our etc.). It will be appropriate to avoid first-person pronouns in article, and will be appropriate to change those section from first-person pronouns to passive voice.

Author Response

Comment 1: The article reports on the calibration technique of three different off-the-shelf wearable sensors to monitor infant leg movement. Authors demonstrated offset correction methods and reproducibility of the result across sensors through calibration. The paper is well written and understandable, although in some parts authors used first-person pronouns (e.g. we, our etc.). It will be appropriate to avoid first-person pronouns in article. Change the language from first-person pronouns to passive voice.

Response 1: First person language removed

Comment 2: Title of the article requires modification as it sounds vague. Current title seems like a statement.  It is, normally, a well-known practice to calibrate the wearable sensors prior to use.

Response 2: Title changed as requested

Comment 3: In Abstract section, last two lines sounds repetitive. Authors should modify that part.

Response 3: Last sentence removed

Comment 4: In Page 5 and Line 206, check the sentence "measurement per each axis."

Response 4: We appreciate the reviewer for pointing this out. The sentence should be referring to table 2, not table 1. It is not modified accordingly. Also, we omitted the word "each".

Comment 5: Does the mounting position of the sensor on the body affects the calibration gain and offset error measurement.

Response 5: No, it should not. If such is true for a sensor, it is a flawed sensor that should not be used.

Comment 6: Table 3 results are not conclusive. Especially for Ax6 sensor result become worsen after calibration and filtered.
I. Why authors choose only 200 counts? What if they go for 1000 or 5000 counts? Whether the proximity of the count results for all three sensors will be similar? How repeatable or reproducible the results will be for higher counts?

Response 6-I: The number of counts can be increased. Extra movements will show acceleration magnitude profiles similar to those of the 200 movements recorded in this report. Therefore, it can be deduced that the proximity of the count results for all three sensors will be similar.

II. For Opal v2 and Active HR2, only calibration does not have any affects. But after offset error count results moves close to the actual reading.
III. Whereas, for Ax6 sensor error increases from 7.5% for 185 counts (raw) to 12.5% after C+F modification. So the question remains: what type of accuracy authors are looking from the sensors? What is the level of tolerance (error in %) for this measurement?

Response 6-II/III: The difference in the number of movement counts after each preprocessing step results from the combination of the amount of error each sensor displays and the work of the threshold-based movement detection algorithm used in the study. The amount of error the reviewer is pointing out is never solely attributed to the limited capacity of one sensor, but also to the threshold calculated from the sensor’s data. Again, the goal of this study is to show how calibrating and correcting for measurement errors can improve the reproducibility of results across sensors when used with a threshold-based detection algorithm, as now clarified in lines 76-78 of 2. Background.

IV. Can authors demonstrate results for actual measurement of the sensors on infant leg movement after calibration and off-set error correction. And then compare the sensors results with the established state-off-art measurement tool/ method like, three-dimensional motion capture systems or video-based movement analysis. This will properly demonstrate if the method really works in operation!

Response 6-IV: We appreciate this comment and agree that the suggested idea will demonstrate the efficacy of our proposed calibration method. We ask the reviewer to understand, however, that capturing infant movements using conventional motion capture systems is more than challenging. Video-based movement analysis is another option that is widely used in developmental studies, but this also requires a lot of time and resources to be carried out. Even coding a five-minute video of infant movement requires several hours from each of a minimum of 2 experienced coders (to report intraclass correlation and make sure that the coded movements are identified by mutual agreement and not by whim). Furthermore, counts obtained from video are necessarily subjective and likely to be less reproducible than those produced by algorithms operating on well-calibrated sensors.

Reviewer 2 Report

Comments and Suggestions for Authors

This work presents a data analysis of measurement error types for uncalibrated wearable sensors, and testing three commercially available sensors for offset, misalignment, gain error, and noise values. The research is meaningful and described in the correct language, but it needs revision before publication in Sensors.

1.            The author needs to introduce the significance and importance of calibration in wearable sensors.

2.            Keywords should be rewritten.

3.            The experimental design section lacks detailed description.

4.            Whether the authors recorded data on leg movements of real infants?

5.            Sufficient references should be added to the manuscript.

6.            Whether the calibration method proposed by the author is universal?

7.            Why did the authors choose these three different commercially available sensors in this manuscript?

Author Response

This work presents a data analysis of measurement error types for uncalibrated wearable sensors, and testing three commercially available sensors for offset, misalignment, gain error, and noise values. The research is meaningful and described in the correct language, but it needs revision before publication in Sensors.

Comment 1: The author needs to introduce the significance and importance of calibration in wearable sensors.

Response 1: This is described in the fourth paragraph of the Introduction. We clarified that researchers “need to understand the kinds of errors to which multi-axis accelerometers are prone and how they might affect the results of movement detection and counting algorithms”. Further, the Background section presents a real example where calibration (error correction) can help with correct movement identification, as now better explained in lines 76-78.

Comment 2: Keywords should be rewritten.

Response 2: We now have added two new keywords: wearable sensors and reproducibility. We think the first keyword shows that our report fits the theme of the special issue (Wearable Sensors for Human Health Monitoring and Analysis). We also believe that the second keyword is an important message this report delivers. We think the previous four keywords (calibration, movement quantification, offset error, inertial measurement unit) are “specific to the article, yet reasonably common within the subject discipline”, as recommended by the “Instructions for Authors” of the journal.

Comment 3: The experimental design section lacks detailed description.

Response 3: We have modified the last paragraph of Section 3.2 (Preparation of datasets). We specified that the sensors were placed on the right dorsal forearm of an adult, with the specific location described with greater details. We also clarified that the arm was moving in the air, not touching any surface.

Comment 4: Whether the authors recorded data on leg movements of real infants?

Response 4: Section 3.2 (Preparation of datasets) originally stated that “a movement dataset was prepared from 5 minutes of recording using all three sensors wrapped together and placed on the forearm of an adult (JO)”

Comment 5: Sufficient references should be added to the manuscript.

Response 5: We have added more references to demonstrate that the sensors (Ax6 and movesense, specifically) selected for the study are widely used in movement quantification research. Previously, only one representative work for each sensor was cited. We now have five in total. In addition, as indicated in item 6, we added two more references to show that the calibration method we introduce has been reviewed by different researchers over time.

Comment 6: Whether the calibration method proposed by the author is universal?

Response 6: We initially cited Lötters et al. (1998) in the introduction (page 2, line 71, reference number 9) to indicate that the method has been introduced before. The sentence is now further supported by articles that either developed this original method (ref # 11) or reviewed different calibration methods (ref # 10).

Comment 7: Why did the authors choose these three different commercially available sensors in this manuscript?

Response 7: The three sensors are among the widely used sensors to measure human movement and physical activity. Each sensor has a considerable number of publications in which the sensor was used, and the representative publications for each sensor are referenced.

Round 2

Reviewer 1 Report

Comments and Suggestions for Authors

Authors have taken into account most of the comments and modified the manuscript. However, the abstract of the article needs some modification. In a short abstract, authors use the word “we” three times.

Comments on the Quality of English Language

Minor editing might be required.